# Dynamic range adaptation in primary motor cortical populations

**Robert G Rasmussen[1,2], Andrew Schwartz[1,2,3], Steven M Chase[2,4]***

[1]Department of Bioengineering, University of Pittsburgh, Pittsburgh, United States; [2]Center for the Neural Basis of Cognition, Carnegie Mellon University, Pittsburgh, United States; [3]Department of Neurobiology, University of Pittsburgh, Pittsburgh, United States; [4]Department of Biomedical Engineering, Carnegie Mellon University, Pittsburgh, United States

**Abstract** Neural populations from various sensory regions demonstrate dynamic range adaptation in response to changes in the statistical distribution of their input stimuli. These adaptations help optimize the transmission of information about sensory inputs. Here, we show a similar effect in the firing rates of primary motor cortical cells. We trained monkeys to operate a brain-computer interface in both two- and three-dimensional virtual environments. We found that neurons in primary motor cortex exhibited a change in the amplitude of their directional tuning curves between the two tasks. We then leveraged the simultaneous nature of the recordings to test several hypotheses about the population-based mechanisms driving these changes and found that the results are most consistent with dynamic range adaptation. Our results demonstrate that dynamic range adaptation is neither limited to sensory regions nor to rescaling of monotonic stimulus intensity tuning curves, but may rather represent a canonical feature of neural encoding.

*For correspondence: schase@cmu.edu

**Competing interests:** The authors declare that no competing interests exist.

## Introduction

The range of natural stimuli a person or animal encounters in a single day is constantly changing. A common example is walking out of a dark movie theater and into a brightly lit lobby. Although the illuminance of light may change over three orders of magnitude between these two rooms, within seconds we are able to see clearly. This ability can be explained by the behavior of neurons that encode information about light intensity. Neurons in primary visual cortex and the retina encode the intensity of light with their firing rates (*Kuffler, 1953*; *Hubel and Wiesel, 1962*). These tuning relationships typically follow monotonic curves between firing rates of zero and the maximum firing rates of the neurons (i.e. the dynamic range of the neurons). The resolution in the relationship is determined by how many steps in the firing rate can be distinguished. The resolution is an information metric and is a function of both the slope of the curve and the amount of noise in the firing rate. When the level of light intensity changes from the dark theater to the bright lobby, the neural tuning curves shift and adjust their slopes to provide resolution over the new range of light intensity (*Shapley and Enroth-Cugell, 1984*; *Ohzawa et al., 1985*; *Smirnakis et al., 1997*; *Ohzawa et al., 1982*; *Harris et al., 2000*). This phenomenon is commonly referred to as dynamic range adaptation. In addition to this example in the visual system, dynamic range adaptation has also been documented in other sensory areas, including the auditory (*Kvale and Schreiner, 2004*; *Ulanovsky et al., 2004*; *Dean et al., 2005*; *Nagel and Doupe, 2006*; *Rabinowitz et al., 2011*) and somatosensory (*Maravall et al., 2007*; *Katz et al., 2006*) systems. This phenomenon has been described as optimizing the transmission of information about input stimuli within the limited tuning ranges of the neurons (*Brenner et al., 2000*; *Fairhall et al., 2001*; *Wark et al., 2007*; *Atick, 2011*).

**eLife digest** Most cameras are equipped with an auto-contrast feature that enables them to take high quality pictures in a wide range of lighting conditions. Auto-contrast works by increasing the sensitivity of the camera to light in dimly lit surroundings, but reducing it in bright conditions to ensure that images do not become saturated. Our visual system is equipped with a similar feature. Neurons in the visual system increase or decrease their sensitivity to light as appropriate to enable us to see in both dimly lit rooms and dazzling sunshine.

This process, which is known as dynamic range adaptation, also occurs in neurons that are sensitive to sound or touch. Rasmussen et al. therefore wondered whether the same might hold true for neurons that encode non-sensory stimuli such as the direction of movement. Would these neurons change their sensitivity to direction if presented with a wide range of possible directions instead of a narrow range? If so, this would suggest that dynamic range adaptation occurs throughout the nervous system.

To find out, Rasmussen et al. trained two rhesus macaque monkeys to use their brain activity to move a cursor on a virtual reality screen in either 2D or 3D. Studying this brain activity showed that neurons became less sensitive to the cursor's direction of movement when the task switched from 2D to 3D. This makes sense because in a 3D task, which also features depth, the neurons have a greater range of possible movement directions to encode. Conversely, the neurons became more sensitive to the direction of movement when the task switched from 3D to 2D. Under these circumstances the neurons can use activity that was previously dedicated to encoding depth to instead represent the 2D space in finer detail.

These results presented by Rasmussen et al. raise several additional questions. Are the mechanisms that support dynamic range adaptation the same in sensory and motor neurons? If these neurons also encode other aspects of movement, such as speed, would these also be included in the same range as direction or is the adaptation process segregated by specific parameter categories? And how do these changes in sensitivity affect the movements that animals produce?

In sensory systems, meaningful input variables can be described concretely. In motor systems, controlled parameters are still being debated. Force was one of the earliest output variables described, and many studies have demonstrated that firing rates of neurons in primary motor cortex (M1) modulate with output force levels in monotonic relationships (*Evarts, 1966*, *1968*; *Humphrey et al., 1970*; *Hepp-Reymond et al., 1978*; *Georgopoulos et al., 1992*). Humphrey et al. briefly noted in their work that the value of regression coefficients relating motor neuronal activity to force scaled with the range of forces in their task (*Humphrey et al., 1970*). This finding could be explained by dynamic range adaptation. As the range of stimuli (output forces) changes, the slope of the tuning curve changes to encode the range of forces within a limited range of firing rates. Hepp-Reymond et al. later explored this phenomenon in greater detail (*Hepp-Reymond et al., 1999*). They demonstrated that motor cortical neurons encoding force during an isometric pinching task undergo dynamic range adaptation with changes in the task context and cued expectation of the task context.

In later work, the direction of arm movement received attention as a controlled variable (*Georgopoulos et al., 1982*, *1988*; *Schwartz et al., 1988*). These studies demonstrated a cosine-tuning relationship between movement direction and neuronal firing rates. Each unit was described as having a preferred direction where the firing rate was maximal for movements in that direction and minimal for movements in the opposite direction. In this classical model, the firing rates linearly modulate between the minimum and maximum rates with the cosine of the angle between the movement direction and preferred direction. This tuning model has been demonstrated for movements in both two- and three-dimensional arm reaches. Notably, the model predicts a lower range in firing rates for 2D movements than 3D movements when the preferred direction is not coplanar with the 2D movement space. This raises the unique question of whether changing the output movement statistics from 3D movements to purely 2D movements would lead to dynamic range adaptation.

Studying these differences with actual arm movements is difficult. Regardless of a 2D or 3D context, arm movements are still within a dynamic environment subject to many external factors including gravity, inertia, and arm kinematics to which neurons may modulate. In an effort to enforce a direct relationship between firing rates and movement direction, we investigated this question using a brain-computer interface (BCI). Essentially, the BCI task allowed us to null out those external factors and purely investigate the differences in tuning between 3D and 2D movements. We trained monkeys to move a cursor on a 3D computer screen under brain control, using either a population vector algorithm (PVA) or an optimal linear estimator (OLE) decoder (*Chase et al., 2009*). The monkeys controlled the cursor by modulating the firing rates of neurons recorded in the motor cortex as they performed a classic center-out task in both 2D and 3D environments. We found that neurons within M1 undergo a change in tuning between the 2D and 3D contexts. We then leveraged the simultaneity of the recordings to test several hypotheses about population-based mechanisms of these changes. We found that the results are consistent with dynamic range adaptation.

## Results

Two male Rhesus macaques implanted with Utah arrays were trained to perform 2D and 3D center-out tasks in virtual reality, under brain control (*Figure 1A,B*). To map the neural activity to movement of the cursor, the activity of each unit was assumed to follow a 3D cosine tuning model. With the fitted models of neural activity, cursor movement was updated online every 33 ms with outputs from the decoders as described in more detail in the Materials and methods section. For the 2D task, the

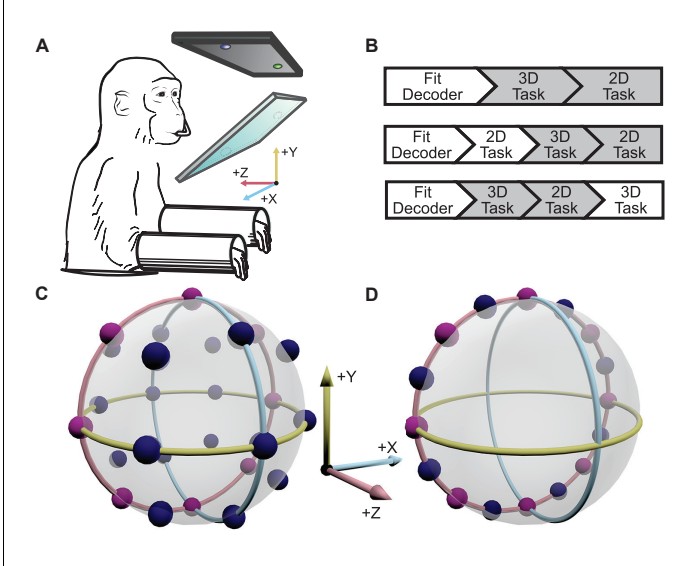

**Figure 1.** Experimental setup. (**A**) The monkey controlled a computer cursor on a stereoscopic monitor viewed through a mirror, using neural signals. (**B**) Each recording session began with a brief calibration session to build the decoder. The decoder was used for a block of 3D trials (between 9 and 13 trials per target), followed by a block of 2D trials (between 15 and 27 trials per target). In some sessions, an additional block of 2D trials would proceed the blocks of 3D and 2D trials, or an additional block of 3D trails would follow the blocks of 3D and 2D trials. We focused our analyses on the first block of 3D trials and the block of 2D trials immediately following it (the gray boxes). In the supplementary materials, we demonstrate that our findings are invariant to the order of the blocks of trials. (**C**) Locations of the 26 targets in the 3D context. (**D**) Locations of the 16 targets in the 2D context. The eight targets that were common to the 2D and 3D contexts are shown in pink. The red, yellow, and light blue hoops represent the *xy*-plane, *xz*-plane, and *yz*-plane, respectively. Data of the target locations are available in *Figure 1—source data 1*.

The following source data is available for figure 1:

**Source data 1.** Target location dataset.

$z$-component of the decoded movement was zeroed to constrain movement to the $xy$-plane. There were 26 targets in the 3D context and 16 targets in the 2D context. The two target sets included eight common targets, allowing for direct comparison of tuning between the two task contexts (*Figure 1C,D*). We analyzed 18 sessions from Monkey A and 11 from Monkey C. Our analysis was focused on the units that contributed to cursor movement for both contexts. The average size of this overlapping population was 26.4 units for Monkey A (min: 13; max: 32) and 37.2 units for Monkey C (min: 30; max: 45). For five recording sessions from Monkey A and 4 from Monkey C, the entire population of units controlling the cursor movement was identical between the two contexts. The average population size for these sessions was 28.6 units for Monkey A (min: 26; max: 32) and 39.5 units for Monkey C (min: 33; max: 45). Six units that appeared to lose signal during the recording session were removed from these analyses (*Figure 2—figure supplement 1*).

## Firing rates change with context

Our first finding is that firing rates during the virtual reaches depend on whether the task context is 2D or 3D. Firing rates for each successful trial were calculated in time windows from target presentation until target acquisition. The rates were averaged across repeated trials to each target, to compute the trial-averaged firing rates. For the set of common targets between contexts, we calculated the average firing rates separately for each context. In *Figure 2AB*, we plot the average firing rates to the targets that lay in the $xy$-plane for both contexts. The blue lines show the trial-averaged firing rates of the neurons to all targets from the 2D context, plotted as a function of target direction (azimuth). The pink lines show the trial-averaged firing rates of the neurons for the eight targets in the $xy$-plane during the 3D context, again as a function of azimuth. The horizontal dotted lines show the minimum and maximum trial-averaged firing rates amongst the entire set of 26 targets during the 3D context. As expected, the azimuthal preferred directions are nearly the same between the two contexts (*Figure 2C*). The changes in azimuthal preferred directions are insignificant under the Moore test of paired samples of angles for both monkeys (*Moore, 1980*; *Zar, 2010*). However, the ranges in the tuning curves are not identical between the two contexts. Instead, the range of firing rates for all 16 targets during the 2D context approaches, and occasionally exceeds, the range observed for all 26 targets during the 3D context. This finding is even more striking when considering the firing rates for the set of eight targets that were identical between contexts. The units show greater maximal firing rates and lower minimal firing rates in the 2D context compared to the 3D context for the BCI task.

To quantify this change, we computed the planar tuning ranges as the difference between the minimum and maximum average firing rate amongst the set of eight identical targets for the two contexts. We found that 334 of 471 units in Monkey A and 230 of 409 units in Monkey C exhibited an increase in planar tuning range in the 2D context relative to the 3D context. The mean ±standard error (SE) increase in tuning range was 4.02 ± 0.33 Hz for Monkey A and 0.81 ± 0.25 Hz for Monkey C (*Figure 2D*). The range in firing rates for a set of identical stimuli changes with the context of the task. For the purposes of comparison, the differences in the tuning ranges for the set of all targets between each context are shown in *Figure 2E*.

In our investigation, we varied the experimental conditions to analyze the stability of the tuning changes. While *Figure 2D* shows the tuning difference when the 2D context follows the 3D context, we found that similar tuning changes occur when the order is reversed (*Figure 2—figure supplement 2A*). We also explored the effect that a different choice in decoder might have on the result. This is important, because the PVA explicitly normalizes every neuron by its modulation depth, while the OLE does not. The findings reported above are results inclusive of all recording sessions regardless of decoders. When we separately looked at sessions using either the PVA decoder or the OLE decoder, we found that the results were individually significant for both decoder types (*Figure 2—figure supplement 2B*). Additionally, in a subset of the recording sessions, the entire population used to control cursor movement was identical between the two contexts. When we looked at the population response from these sessions, we again found similar changes in tuning range (*Figure 2—figure supplement 2C*). Additionally, to test the effects of zeroing out the $z$-dimension, we conducted a series of recording sessions in which we simply re-adapted the decoder for the 2D context rather than use the decoder from the 3D context with the $z$-dimension zeroed out. When we separately looked at sessions using either re-adapted decoders or identical decoders with the $z$-dimension zeroed out in the 2D context, we found that the results were individually significant for both

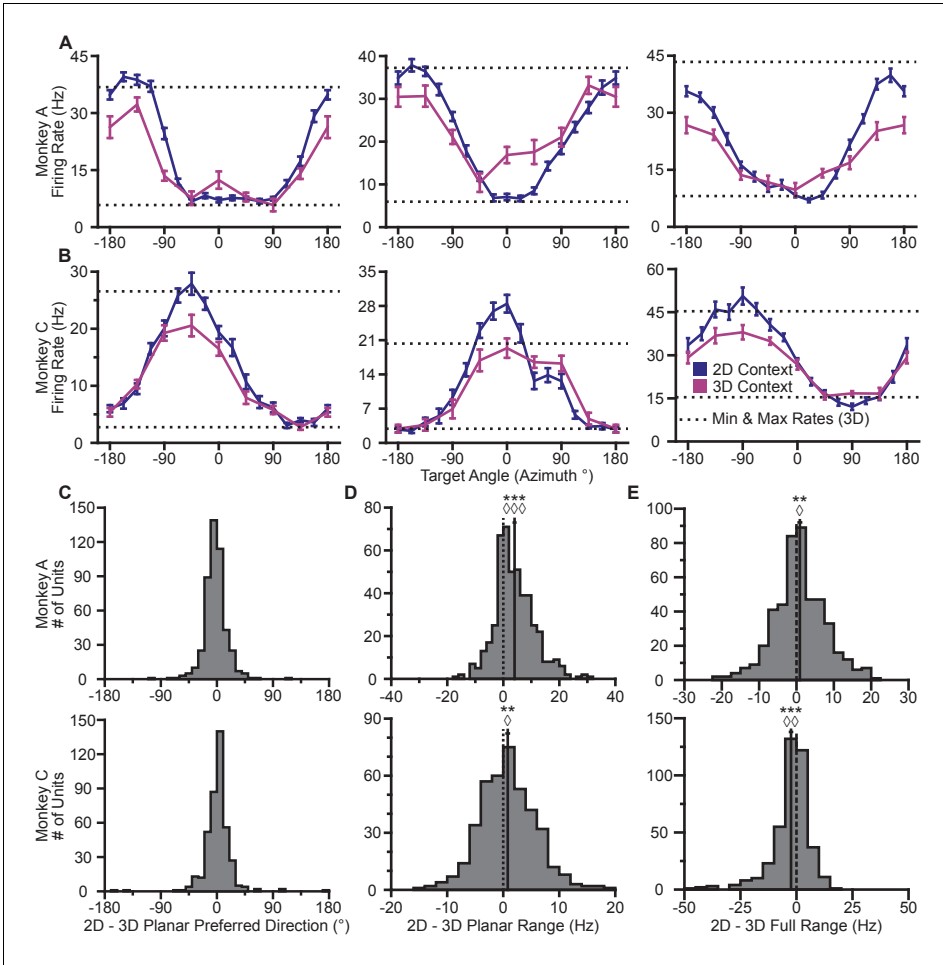

**Figure 2.** Tuning differences between the 2D and 3D contexts. (**A, B**) Selected tuning curves for the targets in the *xy*-plane during the 3D (pink) and 2D (dark blue) contexts from Monkey A (panel **A**) and Monkey C (panel **B**). Firing rates show the mean ± SE for each target. There were 10 trials per target in the 3D task for all units shown with the exception of the last unit in (**B**) where there were 12 trials per target. There were 15 trials per target in the 2D task for all units shown with the exception of the first two units in (panel **A**) where there were 27 trials per target. The dashed horizontal lines show the minimum and maximum target-averaged firing rates for all 26 targets in the 3D context. (**C**) Changes in azimuthal preferred direction for Monkey A (top) and Monkey C (bottom). Firing rates were used to fit log-linear tuning models independently for the 2D and 3D contexts to calculate the preferred directions of the units. The preferred direction from the 2D context was compared to the *x* and *y*-components of the preferred direction from the 3D context to calculate the change in the azimuthal preferred direction. The changes are insignificant under the Moore test of paired samples of angles for both Monkey A ($R' = 0.817$, $n = 471$, $P > 0.1$) and Monkey C ($R' = 0.887$, $n = 409$, $P > 0.05$). (**D**) Histograms of the differences in tuning ranges for the set of eight identical targets between the 2D and 3D contexts for Monkey A (top) and Monkey C (bottom). For both panels: solid vertical lines, means of distributions; solid horizontal lines, mean ± SE; dashed vertical lines, point of equality between the two tuning ranges. In both subjects, the 2D planar dynamic range was larger than the 3D planar dynamic range (Monkey A: ***$P \ll 10^{-20}$, Wilcoxon signed-rank test, $W = 88059$, ◊◊◊$P < 10^{-18}$, sign test, $S = 334$, $n = 471$ units; Monkey C: **$P = 0.0032$, $W = 48975$, ◊$P = 0.0134$, $S = 230$, $n = 409$ units). (**E**) Histograms of the differences in tuning ranges for the full set of targets between the 2D and 3D contexts for Monkey A (top) and Monkey C (bottom). The full tuning range was greater in the 2D context for Monkey A (**$P = 0.0081$, $W = 63402$, ◊$P = 0.0426$, $S = 258$, $n = 471$) and in the 3D context for Monkey C (***$P < 10^{-5}$, $W = 31208$, ◊◊$P = 0.0011$, $S = 171$, $n = 409$). Data to recreate the tuning curves and histograms are available in *Figure 2—source data 1*.

The following source data and figure supplements are available for figure 2:

**Source data 1.** Example tuning curves, preferred directions, and tuning ranges dataset.

*Figure 2 continued*

**Figure supplement 1.** Units removed from analysis due to signal loss.

**Figure supplement 1—source data 1.** Tuning curves of neurons removed from analysis dataset.

**Figure supplement 2.** Invariance of tuning range changes.

**Figure supplement 2—source data 1.** Context order, decoder type, changed population, and re-adapted decoder tuning range dataset.

**Figure supplement 3.** Tuning range changes for Monkey A for one recording session.

**Figure supplement 3—source data 1.** Single session tuning curves dataset.

**Figure supplement 4.** Difference in tuning ranges in single sessions.

**Figure supplement 4—source data 1.** Single session tuning range dataset.

**Figure supplement 5.** Minimal changes with repeated contexts.

**Figure supplement 5—source data 1.** Repeat context tuning range dataset.

types of sessions (*Figure 2—figure supplement 2D*). The changes in tuning ranges are also apparent when looking at firing rates from a single recording session from Monkey A (*Figure 2—figure supplement 3*). The 2D and 3D planar tuning ranges for this recording session and a similar single recording session from Monkey C are compared in *Figure 2—figure supplement 4*.

As shown in *Figure 1B*, there was a subset of recording sessions where a task context was repeated after the alternative context. These sessions allowed us to investigate the stability of the tuning ranges for identical contexts across the duration of a recording session (*Figure 2—figure supplement 5*). We found a slight decrease in the tuning ranges for repeats of the 2D context for Monkey A and repeats of the 3D context for Monkey C: the second presentation tended to have decreased tuning relative to the first. This may be due to fatigue or satiation. Notably, this decrease in tuning on repeated presentations is in the opposite direction as the increase in tuning we observe when moving from the 3D to the 2D context. This suggests that our results from *Figure 2D* may be a slight underestimate of the actual tuning increase.

## Tuning range increases are driven by dynamic range adaptation

We next investigated several mechanisms that might be driving the context-dependent firing rate changes we observed during the BCI task. We considered three possible mechanisms: (1) a re-aiming mechanism, in which subjects aim at points other than the targets (*Jarosiewicz et al., 2008*; *Chase et al., 2010*), (2) a speed gain mechanism, in which subjects change intended speed between the two contexts (*Moran and Schwartz, 1999*), and (3) a dynamic range adaptation mechanism, in which units adapt to utilize their available dynamic range in both contexts (*Brenner et al., 2000*). These strategies each make different predictions about how tuning changes will correlate across populations of simultaneously recorded cells. Below we discuss each of these mechanisms in turn.

### Re-aiming mechanism

This mechanism refers to the process of aiming at a virtual target rather than the one that is presented (*Jarosiewicz et al., 2008*; *Chase et al., 2010*). Under re-aiming, tuning curves are assumed to remain fixed, but firing rates to identical targets can change if their re-aiming points are different across contexts. Re-aiming processes have been found to explain the majority of short-term adaptation to visuomotor rotations experienced under BCI control (*Chase et al., 2012*). Why might re-aiming occur between the 3D and 2D contexts? Under the 3D context, re-aiming would likely result in

movement of the cursor away from the target, and thus would not be beneficial. In the 2D context, however, aiming points anywhere in the $z$-dimension that maintain a constant projection in the $xy$-plane would all result in a straight movement. If the preferred directions of the population are evenly distributed, there would be no speed benefit by either mentally 'pushing' or 'pulling' the cursor above or below the $z$-plane during the 2D task: increased firing rates for the population of neurons on one side of the plane would be balanced by decreased firing rates for the population of neurons on the other side of the plane such that speed stays constant. However, if there are biases in the distributions of preferred directions in depth, re-aiming in depth could increase firing rates more than they decrease, allowing the subject to push the cursor faster toward the displayed target. With this mechanism, units would undergo either an increase or decrease in tuning range, depending on the elevation of the unit's preferred direction relative to the elevation of the re-aiming point.

We illustrate these tuning-dependent changes for an example scenario in *Figure 3A*. The color on the sphere indicates the relative tuning range change for a unit with a preferred direction at that point. The preferred directions of three example units are shown as the light green points on the sphere. Although the three units have nearly the same azimuthal preferred directions (near 45°), the elevation angle of the preferred direction vectors are positive for the first two units and negative for the third. Under the re-aiming mechanism, the re-aiming point for the target with an azimuthal direction of 45° in the 2D context would be above that target, closer to the preferred directions of the first two units and farther from the preferred direction of the third unit. The re-aiming point would lie above the $xy$-plane because of the bias in the distribution of the three units. With this re-aiming point, the first two units would have an increased firing rate and the third unit would decrease its firing rate in the 2D context to the target with an azimuthal direction of 45°. The increased firing of the two units would outweigh the decreased firing rate of the third unit, subsequently pushing the

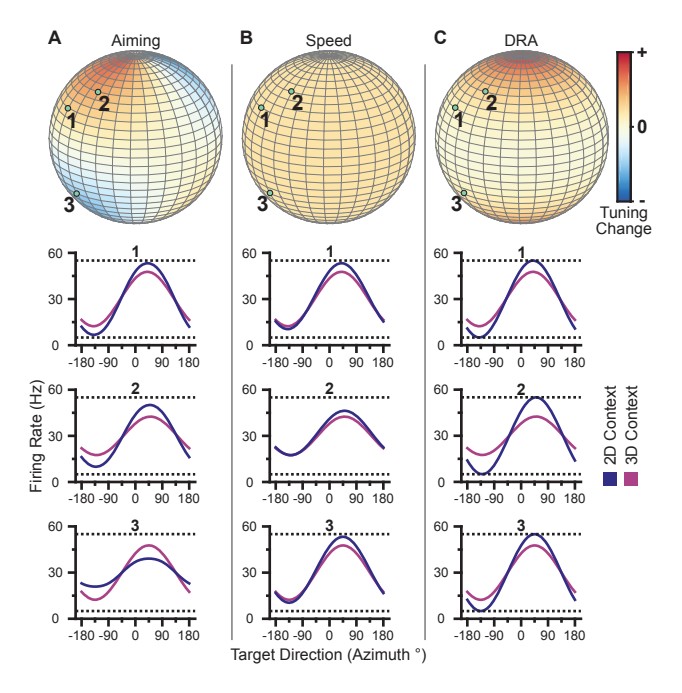

**Figure 3.** Hypotheses for tuning changes. (**A**) Illustration of the re-aiming strategy hypothesis. (**B**) Illustration of the speed hypothesis. (**C**) Illustration of the dynamic range adaptation hypothesis. For each hypothesis, the first row shows heat maps of the relative change in tuning range as a function of preferred direction. For illustrative purposes, the heat map in (panel **A**) is shown as $\frac{\text{2D Range} - \text{3D Planar Range}}{\text{3D Range}}$ and scaled in the range $[-0.25, 0.25]$. The other heat maps are shown as $1 - \frac{\text{3D Planar Range}}{\text{2D Range}}$ and scaled in the range $[-1, 1]$. The bottom three rows show example changes in tuning functions for three units with preferred directions drawn as the light green points on the heat maps.

cursor to the target faster. The tuning curves for these three units for the 3D and 2D contexts under the re-aiming mechanism are drawn under the re-aiming sphere. In this hypothetical example, the first two units demonstrate an increased tuning range in the 2D context, whereas the third unit demonstrates a decreased tuning range.

## Speed gain mechanism

This mechanism refers to the process of encoding different speeds in the firing rates between the two contexts. Moran and Schwartz demonstrated that movement speed can have both additive and multiplicative effects on the firing rate of individual neurons (*Moran and Schwartz, 1999*). They modeled the relationship between movement velocity, $\mathbf{v}$, and neural firing rate, $r_i$, with the following equation:

$$r_i = b_{0,i} + ||\mathbf{v}||b_{s,i} + m_i\mathbf{v} \cdot \mathbf{p}_i \qquad (1)$$

where $||\mathbf{v}||$ is the movement speed and $b_{s,i}$ scales the additive effect of the speed. Unlike the traditional linear tuning model (*Georgopoulos et al., 1982*), the modulation depth, $m_i$, scales not only the directional tuning effects but also the multiplicative effect of speed. The decoding algorithms used in these studies convert the firing rates of the population into cursor velocity. A more detailed description of the decoding algorithms is provided in the Materials and methods section. Briefly, more extreme firing rates (i.e. very low and high firing rates relative to baseline) would increase the speed of the cursor. Thus, increases in intended cursor speed during the 2D context would increase the tuning range of the units. We illustrate the relative change in tuning range as a function of preferred direction predicted by a change in intended speed in *Figure 3B*. Unlike the re-aiming mechanism, the relative change in tuning range expected does not vary with preferred direction. Instead, the relative change in tuning range is identical for all units. Following *Equation 1,* the tuning range for a unit, $\rho_i$, is the difference between the maximum and minimum firing rate, $r_i$, over the range of targets, $\mathbf{d}$, in the $xy$-plane. Because the tuning range is the difference in rates across targets, the additive effect cancels and only the multiplicative effects of speed remain in the tuning range, $\rho_i = 2m_i||\mathbf{v}||\cos(\phi_i)$. Thus, if the intended speed during the 2D context is 20% greater than that in the 3D context, the 2D tuning range will be 20% greater than the 3D planar tuning range.

## Dynamic range adaptation mechanism

This mechanism refers to the process of the units adjusting their tuning curves to utilize their full tuning range in the given task context. This process is akin to the dynamic range adaptation observed in sensory cortices, where neurons adjust their tuning functions to optimally encode the range of input stimuli. In the 2D context, the $z$-dimension is no longer relevant. Subsequently, the portion of the 3D tuning range allocated to encoding the $z$-dimension may be utilized to encode targets in the $xy$-plane during the 2D context. We illustrate the relative change in tuning range as a function of preferred direction predicted by dynamic range adaptation in *Figure 3C*. Unlike a change in intended speed, the relative change in tuning range expected due to dynamic range adaptation varies with the elevation angle, $\phi$, of the preferred direction. Under dynamic range adaptation, the 2D tuning range would be equal to the full tuning range ($2m_i$ from *Equation 1*), whereas the 3D planar tuning range is scaled by the cosine of the elevation angle, $2m_i\cos(\phi_i)$. We refer to this as a 'dose-response' effect. Increasing the 'dose' of the elevation angle leads to an increased response of tuning range change.

We find our data to be most consistent with the dynamic range adaptation mechanism. Two pieces of evidence suggest this. First, our data exhibit the dose-response effect predicted only by dynamic range adaptation. Second, we find no evidence of coordinated, opposing changes in dynamic range for neurons whose 3D preferred directions were mirrored across the $xy$-plane, as predicted by the re-aiming mechanism. We explain these analyses in full below.

To investigate whether the data exhibit a dose-response effect, we plotted the ratio of the 3D planar to 2D ranges against the absolute value of the elevation angle (*Figure 4A,B*). When we fit a weighted linear regression to these data, we found significant, negative slopes, indicating that neurons tuned to larger elevation angles show larger dynamic range changes between 2D and 3D (Monkey A: $P < 10^{-12}$, F-test; Monkey C, $P = 0.00125$). Notably, the ratio of the tuning ranges would be expected to follow a cosine relationship with the elevation angle rather than a linear one. However,

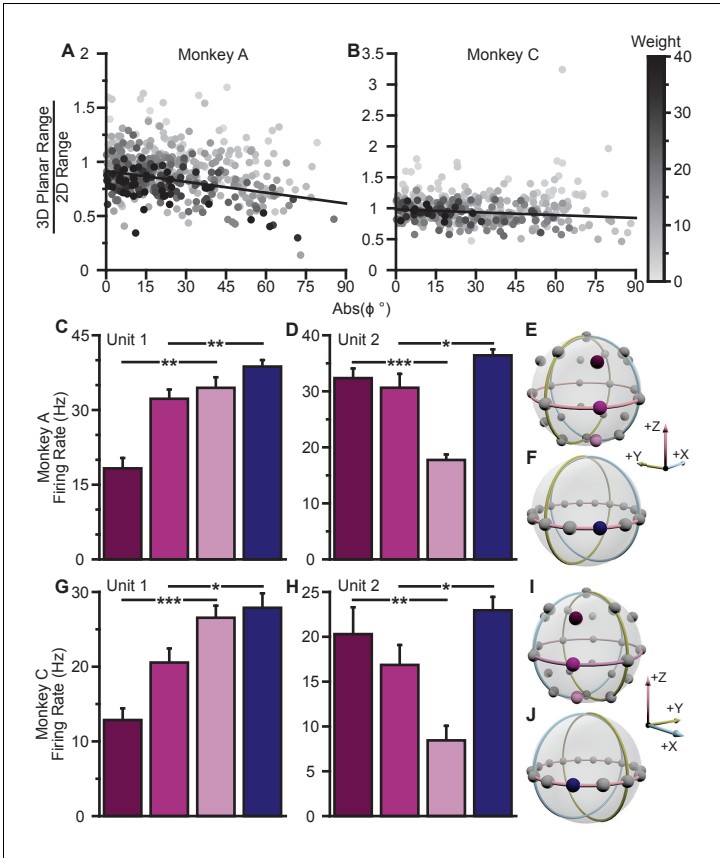

**Figure 4.** Evidence for dynamic range adaptation. (**A**) Dose-response effects in the tuning changes for Monkey A. (**B**) Dose-response effects in the tuning changes for Monkey C. The ratio of 3D planar to 2D dynamic ranges are plotted against the absolute value of the elevation angle ($\phi$) of the PD vectors. The weighted linear fits for the data points are shown in black. The fits have negative slopes. $b_A = -0.0033$, $F_A = 53.9$, $P<10^{-12}$, $n_A = 471$, $b_C = -0.0015$, $F_C = 10.6$, $P = 0.00125$, $n_C = 409$. $b$, slope, $F$, $F$-test versus constant model, $n$, number of analyzed units. The darkness of the points corresponds to the weights used for the linear fits with the legend shown to the right. To provide contrast, the upper threshold of colormap for the weights was set at 40. The tuning curves for the two points that lie significantly off the trend line in (**B**) are shown in **Figure 4—figure supplement 1B**. (**C,D**) Firing rates from Monkey A for the first two units from **Figure 2A**. Both units have an azimuthal PD near $-135°$. The first three bars show the firing rates in the 3D context for the target that lay at $-135°$ on the $xy$-plane (pink) and the two targets immediately above (dark pink), and below (light pink) that target (**E**). The firing rate in the 2D context to the target at $-135°$ is shown in the fourth bar (**F**). Unit 1 had a preferred direction below the $xy$-plane ($U_{1;Z} = 61$, **$P = 0.001$). Unit 2 had a preferred direction above the $xy$-plane ($U_{2;Z} = 154$, ***$P<10^{-3}$). Despite this difference, both units showed an increased firing rate to the planar target in the 2D context compared to the 3D context ($U_{1;2D,3D} = 114$, **$P = 0.0098$, $U_{2;2D,3D} = 126$, *$P = 0.0299$). (**G,H**) Similar set of firing rates from Monkey C for the first two units from **Figure 2B**. The corresponding targets are shown in (**I** and **J**). Both units had an azimuthal PD near $-45°$. Unit 1 had a preferred direction below the $xy$-plane $U_{1;Z} = 69$, ***$P<10^{-3}$). Unit 2 had a preferred direction above the $xy$-plane ($U_{2;Z} = 164$, **$P = 0.0028$). Despite this difference, both units showed an increased firing rate to the planar target in the 2D context compared to the 3D context ($U_{1;2D,3D} = 89$, *$P = 0.0165$, $U_{2;2D,3D} = 95$, *$P = 0.0374$). The firing rates are plotted as the mean $\pm$ SE ($U$, Mann-Whitney $U$ tests, $n_{2D}$ and $n_{3D}$ trials are the same from **Figure 2A,B**). Data to recreate these plots are available in **Figure 4—source data 1**.

The following source data and figure supplements are available for figure 4:

**Source data 1.** Tuning range change versus elevation angles and paired neuron changes dataset.

**Figure supplement 1.** Dose-response effects in firing rate changes.

*Figure 4 continued on next page*

when we repeated this analysis using the cosine of the elevation angle rather than the absolute value, the results did not change (*Figure 4—figure supplement 1A*). These results are only predicted by the dynamic range adaptation hypothesis: units appear to increase their range of modulation to include the portion of the tuning range that encoded and controlled the *z*-dimension in the 3D context. Thus, this changes their contribution to the encoding and control of the *x* and *y*-dimensions relevant in the 2D context of the BCI task. When we compared the full tuning ranges from the 2D and 3D contexts, we found that more than half of the units (54.8%) had greater full tuning range in 2D than 3D for Monkey A, whereas less than half of the units (41.8%) had greater full tuning range in 2D than 3D for Monkey C (*Figure 2E*).

We also searched for evidence of coordinated, opposing changes in dynamic range for scenarios like Units 1 and 3 illustrated in *Figure 3A*, in which a pair of units with nearly identical 2D preferred directions had 3D preferred directions that were mirrored across the *xy*-plane. As explained above, the re-aiming strategy predicts that one unit would show an increased firing rate to the planar target in the 2D context compared to the 3D context, while the other unit would show a decreased firing rate. In the alternative hypotheses, both units would show increased firing rates to the planar target in the 2D context. Results from a pair of units matching this scenario are shown for Monkey A (*Figure 4C,D*) and Monkey C (*Figure 4G, H*). Activity for these example pairs of units is inconsistent with the re-aiming strategy. Both units demonstrate maximal firing rates to the planar target in the 2D context. To study whether the results were consistent with the re-aiming strategy at the population level, we developed an algorithm to predict the optimal aiming points for the population in the 2D context, based upon the firing rates in the 3D context. This algorithm allows us to predict which neurons would show increases in dynamic range under the re-aiming mechanism, and which neurons would show decreases in dynamic range. Note that under both the speed gain and dynamic range adaptation mechanisms, no neurons are predicted to decrease their dynamic range. We applied this analysis to the recording sessions where the entire population was identical between the two contexts. For Monkey A, of the 87 units predicted to increase tuning range, 66 were observed to increase. Of the 55 units predicted to decrease tuning range, only eight were observed to do so. For Monkey C, of the 86 units predicted to increase tuning range, 57 were observed to increase. Of the 72 units predicted to decrease tuning range, only 30 were observed to do so. In both monkeys, more than half of the neurons predicted to show decreases in dynamic range from the 3D context to the 2D under the re-aiming mechanism actually showed

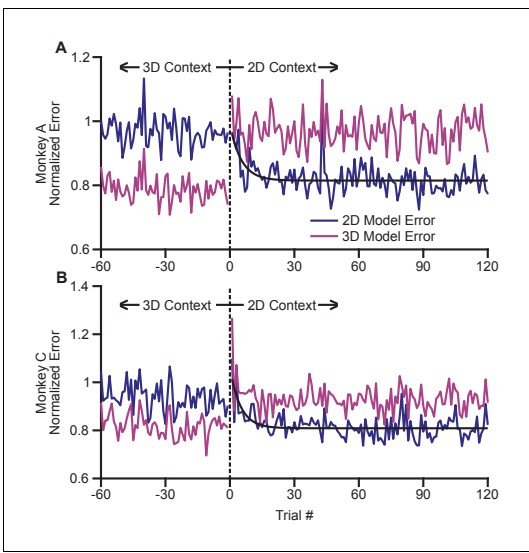

**Figure 5.** Time course of adaptation. The two panels show the time courses of adaptation for Monkey A (**A**) and Monkey B (**B**). The ordinate axes in the plots show the standardized error between the firing rates in the population and the expected response from tuning fits. Each plot shows two different errors: the error based on tuning models from the 2D context (dark blue) and from the 3D context (pink). These standardized errors are averaged across multiple recording sessions for Monkey A (18) and Monkey C (11). The abscissa in the plots show the trial number before and after the change from the 3D context to the 2D context. The data points to the right of zero represent the standardized error for the first 120 trials in the 2D context. The data points to the left of zero represent the last 60 planar target trials of the the 3D context. The dashed black vertical line at zero represents the system down-time while the context was switched. We also plotted exponential fits (black) to the 2D tuning model error for the first 120 trials of the 2D context. The data were fit to the equation: $y = a + b \exp cx$. For Monkey A, the fit parameters (with 95% confidence bounds) were: $a = 0.815(0.807, 0.8233)$; $b = 0.176(0.087, 0.264)$; $c = -0.189(-0.317, -0.060)$; $r^2 = 0.229$. For Monkey C, the fit parameters were: $a = 0.809$ $(0.800, 0.818)$; $b = 0.250(0.150, 0.348)$; $c = -0.202$ $(-0.309, -0.095)$; $r^2 = 0.328$. Data to recreate these plots is available in *Figure 5—source data 1*.

The following source data is available for figure 5:

**Source data 1.** Trial-by-trial tuning model error dataset.

increases in dynamic range, as predicted by the dynamic range adaptation mechanism.

## Other properties of this effect

The adaptation process appeared to occur rapidly with the switch from the 3D to 2D context. In order to investigate the timecourse of this process, we devised a method to estimate changes in dynamic range on a trial-by-trial basis (see Materials and methods). Briefly, we fit cosine tuning models from trials in both the 2D and 3D contexts. These models represent a prediction of the expected population response for each context that can be assessed on a trial-by-trial basis. By investigating the error in the fit of each model when the subject experiences a transition between 3D and 2D, we can assess how quickly the population switches from its expected 3D response to its expected 2D response. To mitigate noise, we normalized the expected error for each neuron by the average residual error for the 2D and 3D contexts, and we averaged across all neurons and across the multiple recording sessions for each monkey. Note that all errors were cross-validated: we fit separate models for even and odd trials, and used them to predict the results on odd and even trials, respectively. We estimated the error for the first 120 trials in the 2D context as well as the last 60 planar target trials from the preceding 3D context. These normalized, averaged errors are shown in *Figure 5*. The plots demonstrate a very rapid shift in the tuning of the population as the context changes from 3D to 2D. Immediately on transition from 3D to 2D (on trial 1), the 3D model error grows considerably. Presumably, the lack of movement in the *z*-dimension cues the subject to the context change. The 2D model error then reduces. We fit exponential models to the reduction in 2D model error for each monkey, and found exponential decay coefficients (with 95% confidence bounds) of 5.3 (3.2, 16.7) trials for Monkey A and 4.6 (3.2, 10.5) trials for Monkey C. Because each trial takes approximately 1 s to perform, the retuning effect nears completion within seconds. The 2D model error is immediately lower than the 3D model error on the first 2D trial and rapidly reduces to a baseline state within the first few trials.

We also assessed the completeness of the dynamic range adaptation. To be complete, the dynamic range in the 2D context should equal the full dynamic range in the 3D context, when all targets are taken into account. We developed the following metric to quantify the completeness of the adaptation:

$$\text{Adaptation} = \frac{\rho_{2D} - \rho_{3D}}{\rho_{3D}^{full} - \rho_{3D}} \qquad (2)$$

Here, $\rho$ represents the tuning range under a specified context and target set. The subscript denotes the context. The superscript 'full' denotes the entire set of targets for a given context, whereas the lack of a superscript represents the set of targets identical between contexts. The terms used in computing this metric are described in greater detail in the Tuning Range Analysis subsection of the Methods. This metric normalizes the change in tuning range such that no adaptation would lie at zero and complete adaptation would lie at one. This metric is problematic for neurons where the elevation angle is near the *xy*-plane and the difference between the full and planar tuning ranges approaches zero. In order to avoid this possible complication, we limited our analysis to those neurons where both the elevation angle was greater than 30° and the denominator was greater than 2 Hz. The resulting histograms from the metrics for Monkeys A and C are shown in *Figure 6*. For

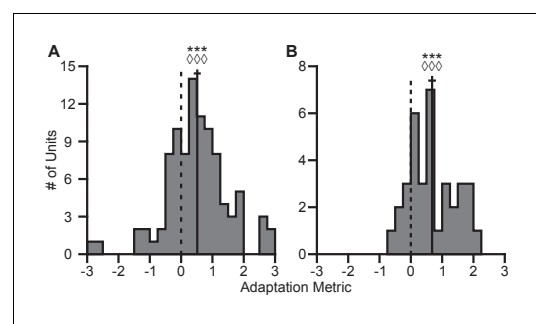

**Figure 6.** Completeness of adaptation. The two panels show histograms of the completeness metric for Monkey A (**A**) and Monkey C (**B**). For both panels: solid vertical lines, means of distributions; solid horizontal lines, mean ± SE; dashed vertical lines, zero point representing no dynamic range adaptation. For both monkeys, histograms were distributed significantly away from the zero point (Monkey A: ***$P<10^{-5}$, Wilcoxon signed-rank test, $W = 3580$, ◊◊◊$P<10^{-3}$, sign test, $S = 68$, $n = 95$ units; Monkey C: ***$P<10^{-4}$, $W = 570$, ◊◊◊$P<10^{-3}$, $S = 29$, $n = 35$ units). The means (± SE) of the metrics were 0.5157 (± 0.1121) for Monkey A and 0.6793 (± 0.1233) for Monkey C. Data to recreate these plots is available in *Figure 6—source data 1*. DOI: 10.7554/eLife.21409.024

The following source data is available for figure 6:

**Source data 1.** Adaptation metric dataset. DOI: 10.7554/eLife.21409.025

both monkeys, the histograms are distributed significantly away from zero under both a sign test and a Wilcoxon signed-rank test. Furthermore, the means ($\pm$SE) of the metrics are 0.5157 ($\pm$0.1121) for Monkey A and 0.6793 ($\pm$0.1233) for Monkey C. These results suggest that the dynamic range adaptation is more than halfway complete for both monkeys. It is worthwhile to note that dynamic range adaptation is not always complete, even in sensory systems. In the auditory system, Dean and colleagues showed a range of retuning abilities in their example neurons (cf. *Figure 1* of *Dean et al. (2005)*). Ohzawa developed an adaptability index to characterize the contrast gain adaptation of cells in visual cortex, and reported values in line with our results (cf. *Figure 5* of *Ohzawa et al. (1985)*).

## No adaptation with hand-control task

BCI provides a powerful framework for addressing basic scientific questions about motor control, including the dynamic range adaptation effects observed here, because the effect each neuron has on cursor movement is fully specified by the experimenter, and because we can obviate complicating factors such as gravity and inertia (*Golub et al., 2016*). However, it is interesting to compare the BCI results to a similar task performed in hand-control. We had a third monkey (Monkey N) perform a center-out reaching task with the cursor controlled by the position of a 3D tracking marker placed on the monkey's hand. For the 2D movements, the z-position of the marker was zeroed out so that the depth of the marker had no effect on the cursor. We computed the tuning ranges for the directionally tuned neurons amongst the set of identical targets for each context (*Figure 7*). Unlike with the BCI task, we see no evidence of dynamic range adaptation in the hand-control version of this task. We speculate on the reasons for this in the Discussion section.

## Discussion

Neurons in sensory regions encode and transmit information about the external environment (*Borst and Theunissen, 1999*). Sensory neurons also undergo dynamic range adaptation with contextual changes in the environment (*Ohzawa et al., 1985*; *Dean et al., 2005*). In our study, M1 neurons demonstrate significant changes in their firing rates to identical targets, when the task context switches from 3D to 2D. These changes are widespread across the recorded population. The observed changes are not consistent with a re-aiming strategy during the 2D context. Furthermore, the observed changes are not consistent with changes in intended speed between the two contexts. Instead, the observed changes appear to be consistent with dynamic range adaptation.

During the 2D context, the z-dimension gets added to the task-irrelevant null space. Without the observed adaptation, the portion of the dynamic range encoding the z-dimension during the 3D context would not be utilized during the 2D context. Our results showed that units begin to utilize that portion of the dynamic range to encode targets in the xy-plane as the task context switches from 3D to 2D. These results are markedly similar to the dynamic range adaptations that occur in sensory regions. Just as neurons in the retina and visual cortex adjust their tuning to encode the current range of light intensities, the recorded units adjusted their

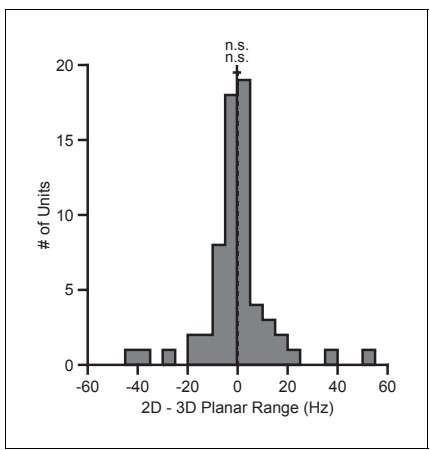

**Figure 7.** Tuning range in hand-control task. The plot shows the histogram of the tuning range difference between the 2D and 3D contexts for the planar targets in the hand-control task. For the plot: solid vertical line, mean of distribution; solid horizontal line, mean $\pm$ SE; dashed vertical line, point of equality between the two tuning ranges. For Monkey N, the tuning ranges were not significantly different between the 2D and 3D contexts (n.s.$P = 0.6349$, Wilcoxon signed-rank test, $W = 969$, n.s.$P = 0.9007$, $S = 31$, $n = 64$ units). Data to recreate these plots are available in *Figure 7—source data 1*.

The following source data is available for figure 7:

**Source data 1.** Hand-control task tuning range dataset.

tuning to encode the current range of targets in the given context of the BCI task.

Researchers have described dynamic range adaptation in sensory systems as functioning to optimize information throughput of the neural populations (*Brenner et al., 2000*; *Fairhall et al., 2001*; *Wark et al., 2007*). The same process appears to be functioning here. Just as the slopes of tuning curves in the visual system increase to transmit more information about a narrower range of light intensities, the slopes of the tuning curves of the M1 units increase to transmit more information about a narrower range of target directions.

Other studies have noted tuning changes of individual neurons during BCI learning tasks (*Jarosiewicz et al., 2008*; *Ganguly and Carmena, 2009*; *Ganguly et al., 2011*; *Chase et al., 2012*). However, the dynamic range adaptation we see appears to be fundamentally different from the effects shown in those studies. First, the time scale of dynamic range adaptation is on the order of seconds (*Figure 5*). The time scale of BCI-learning-related changes in neural tuning is on the order of tens of minutes (*Jarosiewicz et al., 2008*; *Chase et al., 2012*) to days (*Ganguly and Carmena, 2009*; *Ganguly et al., 2011*). Second, the dynamic range adaptation is restricted to changes in tuning depth; changes in preferred direction are minimal. In BCI learning studies, preferred directions typically change as well. Third, dynamic range adaptation occurs without any overt learning pressure. In studies of BCI learning, the mapping between neural activity and cursor movements is changed in such a way to impair task performance. That was not the case here: if the subject continued using the same firing rates in the 3D and 2D contexts, performance would have been unaffected. For these reasons, we speculate that dynamic range adaptation represents an additional degree of flexibility in the responses of neurons in the motor system.

While we observed consistent dynamic range adaptation between the 2D and 3D contexts in the BCI task, we did not observe it in the hand-control version of this task. There are a number of possible reasons for this. In hand control, hundreds of thousands of neurons likely contribute to arm movements, and the overall change in dynamic range might be distributed over all of them, making them difficult to observe. In BCI, only the recorded neurons contribute to cursor movement, which could potentially make the effect easier to see. Another possibility is that the number of movement parameters that an individual neuron tunes for in hand control is quite large, and when moving from 3D to 2D, the percentage change in number of control parameters is small. In BCI, each neuron contributes to only 3D or 2D movements of the cursor, which could also make the effect easier to observe. A third intriguing possibility is that hand control and BCI movements invoke fundamentally different control schemes. The Hepp-Reymond study noted a rescaling of tuning when the output range of required forces was changed (*Hepp-Reymond et al., 1999*). That study was performed under isometric conditions, that is, movements were not physically required. Humphrey and colleagues had subjects perform flexion/extension movements of the wrist (*Humphrey et al., 1970*) under various load conditions, and noted that regression coefficients for decoding scaled with the load. Here, we used a BCI task in which subjects were not physically required to move, and our subjects tend to exhibit very little physical movement during these tasks. In contrast, the hand-control version of this task required physical arm movements, but not variations in output force. It is possible that dynamic range adaptation in the motor system only occurs under conditions of changing force. If this is true, it suggests that isometric force production is a closer analog to BCI than the overt reaching movements to which it is typically compared.

Direction as an important component of the control signal is undeniable. Although directional tuning has been shown to change in a context-dependent manner, it should be emphasized that directional tuning is stable across repetitions within the same condition. Tuning function changes of preferred direction and modulation depth have been observed between brain control and hand control (*Taylor et al., 2002*; *Carmena et al., 2003*; *Lebedev et al., 2005*), between movements in different speed ranges (*Moran and Schwartz, 1999*; *Churchland and Shenoy, 2007*), and between movements in different postures (*Scott and Kalaska, 1995*; *Sergio and Kalaska, 2003*). Tuning can change during learning, when adapting to visuomotor perturbations (*Wise et al., 1998*; *Paz et al., 2003*; *Mandelblat-Cerf et al., 2009*), force field perturbations (*Gandolfo et al., 2000*; *Arce et al., 2010*), BCI decoder changes (*Jarosiewicz et al., 2008*; *Ganguly and Carmena, 2009*; *Hwang et al., 2013*), or even the consistent coupling of sensory cues with rewarded actions (*Zach et al., 2008*). Here, we find another example of tuning change: dynamic range adaptation. One feature that distinguishes dynamic range adaptation from these other sorts of tuning changes is that, using a principle of information processing (namely, that the total range of firing is rescaled according to the output

statistics), the tuning curves from the 2D context may be predicted from the 3D context. Certain types of dynamic range adaptation, such as contrast gain normalization in the visual system, have been linked to a divisive normalization mechanism (*Carandini et al., 1997*). By better defining which aspects of behavior lead to tuning changes and from more complete descriptions of the factors driving these changes, we will gain a deeper understanding of the neural operations taking place during the control of movement.

These results support the hypothesis that neurons in M1 serve to efficiently transmit contextually encoded information. Motor learning in dynamic environments can be viewed as learning efficient ways to encode information. Changes in the environmental context, such as tool use, can lead to changes in the association between M1 activity and motor output (*Quallo et al., 2012*). Even during typical reaches in a static environment, M1 neurons efficiently encode information about the task. The common linear tuning function (*Georgopoulos et al., 1982*) maximizes the output entropy for a uniform distribution of target directions in a 3D environment. This 'histogram equalization' maximizes the information transmission in the low-noise limit (*Nadal and Parga, 1994*; *Laughlin, 1981*). In our studies, the units in M1 have limited dynamic ranges with which to encode cursor movement under the different contexts. Rather than using a subset of the available tuning ranges, the population adaptively displays new coding schemes that fully utilize the ranges of the units comprising the population. This adaptive learning of the M1 population allows for the efficient encoding of task relevant information in changing environments.

## Materials and methods

### Behavioral paradigm and recording

The behavioral training and decoding methods closely followed those described by *Jarosiewicz et al. (2008)*. Briefly, two male Rhesus monkeys were trained to perform a center-out reaching task in a 3D virtual environment. The monkeys sat in a primate chair facing a mirror that reflects a 3D image from a stereoscopic computer monitor. Before electrode implantation, the monkey was trained to move its hand, fitted with an optical marker, to move a cursor from the center of an imaginary cube to targets at lying at one of the eight corners for a liquid reward (*Reina et al., 2001*).

Once the monkeys became proficient at the 3D center-out reaching task, they were implanted with chronic recording arrays. Both monkeys were implanted with a 96-channel array (Cyberkinetics Neurotechnology Systems, Inc.). The implantations were visually placed in the proximal arm area of primary motor cortex. Recordings were amplified, filtered, and sorted on-line with a 96-channel Plexon MAP system (Plexon Inc.). Each monkey was trained to perform the center-out task with both arms restrained by modulating the spiking activity of the recorded units to control cursor velocity. Both monkeys became proficient at the brain-control center-out task in both the 2D and 3D contexts from training sessions prior to starting this study.

Each trial began with the monkey holding the cursor at the task space origin for a randomized period (50–160 ms). The target was presented after this period. The task required the monkey to reach the target within the 2 s reach period and hold the cursor at the target position for a randomized period (0–100 ms). The cursor was then automatically returned to the home position for the start of the next trial. A water reward was delivered for successful trials. The targets were placed at a distance of 85 mm from the center. The cursor and targets' radii were 8 mm each. All procedures were performed in accordance with the guidelines of the University of Pittsburgh's Institutional Animal Care and Use Committee.

### Decoding algorithm

Each recording session began with a calibration brain control session. The calibration session followed the methods described by *Chase et al. (2009)*. The decoding parameters were initially randomized. Eight targets from the 3D context (those lying on the corners of a cube) were then presented, individually in random order, and remained until the end of the movement period. Although the cursor did not move much during the movement period due to the randomized parameters, the firing rates were still modulated to target presentation. Once a cycle set comprising one presentation of each of the eight targets was completed, the firing rates were regressed using the

regress function in Matlab (Mathworks, Inc., Natick, MA) against the target direction according to the linear tuning model:

$$r_i = b_{0,i} + m_i \mathbf{d} \cdot \mathbf{p}_i \tag{3}$$

In a subset of recording sessions, separate calibration sessions were conducted for the 2D and 3D trials. These calibration sessions were conducted immediately prior to the start of the respective context block. The calibration session for the 2D trials differed from the previously described calibration methods only in that the target set comprised 8 targets spaced equally around a circle centered in the $xy$-plane. For all other recording sessions, the decoding parameters derived from the initial calibration session were used for the entirety of the recording session.

Linear decoders (PVA and OLE) were used to decode neural signals into cursor movement. The recorded population was assumed to follow the linear tuning function described in *Equation 3* For each unit, $i$, spike counts were measured at 30 Hz and converted into firing rates, $r_i$, by dividing by the sampling interval. Rates were averaged across the last five time bins and normalized by subtracting the fit baseline firing rate and dividing by the fit modulation depth:

$$u_i = \frac{r_i - b_{0,i}}{m_i} \tag{4}$$

The normalized rates of the $N$ units comprising the population were grouped in the vector $\mathbf{u} = [u_1, u_2, \ldots, u_N]^\top$. Similarly, the preferred direction vectors of the $N$ units were grouped in the matrix $P = [\mathbf{p}_1, \mathbf{p}_2, \ldots, \mathbf{p}_N]$. For the PVA decoder, normalized rates, $\mathbf{u}$, were converted to cursor velocity, $\mathbf{v}$, as:

$$\mathbf{v} = k_s \frac{n_D}{N} P^\top \mathbf{u} \tag{5}$$

where $n_D$ is the number of movement dimensions (either 2 or 3) and $k_s$ is the speed factor to convert the normalized speed to a physical speed in mm/s (values used in our study ranged from 65 to 120). For the OLE decoders, the normalized rates were converted to cursor velocity as:

$$\mathbf{v} = k_s \left(B^\top B\right)^{-1} B^\top \left(\mathbf{r_i} - \mathbf{b_{0,i}}\right) \tag{6}$$

where $B = [m_1 \mathbf{p}_1, m_2 \mathbf{p}_2, \ldots, m_N \mathbf{p}_N]$. Cursor position, $\mathbf{c}$, was updated every sampling interval as:

$$\mathbf{c}(t) = \mathbf{c}(t - \Delta t) + \Delta t \mathbf{v}(t) \tag{7}$$

For trials in the 2D context, the $z$-component of the decoded velocity was set to zero to constrain movement to the $xy$-plane.

## Tuning range analysis

The analyses in this work focuses on the firing rates of the recorded units, $r_i$. These rates are functions of many different variables: the task context (i.e. number of movement dimensions, $n_D$), the target direction ($\mathbf{d}$), the trial number for the target ($j$), and the time points within the trial ($t$). For a given monkey and recording session, the rates of the $N$ units comprising the population were grouped in the vector $\mathbf{r} = [r_1, r_2, \ldots, r_N]^\top$. In this work, we analyzed the average firing rate over the trial duration, $\langle \mathbf{r} \rangle_t$. These time-averaged firing rates were also averaged over the trials for a given context and target, $\langle \mathbf{r} \rangle_{t,j}$. These trial- and time-averaged firing rates remain a function of the task context and target, $\mathbf{f}(n_D, \mathbf{d}) = \langle \mathbf{r} \rangle_{t,j}$. We defined the target set, $\mathbb{A}$, as the target set in the 3D context, $\mathbb{A} = \{\mathbf{d} : n_D = 3\}$. This set comprised 26 targets, $|\mathbb{A}| = 26$. We defined the target set, $\mathbb{B}$, as the target set in the 2D context, $\mathbb{B} = \{\mathbf{d} : n_D = 2\}$. This set comprised 16 targets, $|\mathbb{B}| = 16$. There were eight targets which were common to the two sets, $|\mathbb{A} \cap \mathbb{B}| = 8$. We defined that set intersection as the set of common targets, $\mathbb{C} = \mathbb{A} \cap \mathbb{B}$. The 3D planar range for the units in the population was defined as:

$$\rho_{3D} = \max\{\mathbf{f}(n_D, \mathbf{d}) : n_D = 3, \mathbf{d} \in \mathbb{C}\} \\ - \min\{\mathbf{f}(n_D, \mathbf{d}) : n_D = 3, \mathbf{d} \in \mathbb{C}\} \tag{8}$$

Although all targets in the 2D context lie in the *xy*-plane, we calculated the 2D range over the set of common targets to directly compare the ranges:

$$\rho_{2D} = \max\{\mathbf{f}(n_D, \mathbf{d}) : n_D = 2, \mathbf{d} \in \mathbb{C}\} \\ - \min\{\mathbf{f}(n_D, \mathbf{d}) : n_D = 2, \mathbf{d} \in \mathbb{C}\} \tag{9}$$

Our analysis compared the difference between the 3D planar range ($\rho_{3D}$) and the 2D range ($\rho_{2D}$) for the units comprising the population. For comparisons of the full tuning ranges, the 2D full range was calculated over the entire set of 16 2D targets, $\mathbf{d} \in \mathbb{B}$:

$$\rho_{2D}^{full} = \max\{\mathbf{f}(n_D, \mathbf{d}) : n_D = 2, \mathbf{d} \in \mathbb{B}\} \\ - \min\{\mathbf{f}(n_D, \mathbf{d}) : n_D = 2, \mathbf{d} \in \mathbb{B}\} \tag{10}$$

Similarly, the 3D full range was calculated over the entire set of 26 3D targets, $\mathbf{d} \in (A)$:

$$\rho_{3D}^{full} = \max\{\mathbf{f}(n_D, \mathbf{d}) : n_D = 3, \mathbf{d} \in \mathbb{A}\} \\ - \min\{\mathbf{f}(n_D, \mathbf{d}) : n_D = 3, \mathbf{d} \in \mathbb{A}\} \tag{11}$$

## Optimal re-aiming points

In order to test a re-aiming hypothesis that would explain the change in observed tuning ranges, we devised a method to estimate the optimal re-aiming points for that hypothesis. For each recording session that used the identical set of units to control the BCI task between contexts, the firing rate data from brain control in the 3D context between target presentation and acquisition was used to fit a log-linear tuning model (*Koyama et al., 2010*) to each unit in the population. The log-linear tuning model is similar to the linear tuning model in *Equation 3* with the exception of the exponential relationship in the log-linear model:

$$r_{ll,i}(\mathbf{d}) = \exp(b_{ll,0,i} + m_{ll,i}\mathbf{d} \cdot \mathbf{p}_{ll,i}) \tag{12}$$

The subscripts, *ll*, are used to distinguish the equation and its tuning parameters from those used in the linear tuning model. The tuning parameters were fit for each neuron using the glmfit function in Matlab with the distribution and link parameters set as Poisson and log, respectively. For each of the eight shared targets, an optimal re-aiming point for the 2D context was estimated using the log-linear tuning models and the decoders from each recording session. The optimization was run in Matlab as a constrained minimization to move the cursor closest to the target after one time-step. The optimization was constrained to ensure that the single re-aiming point for each target and recording session lies in $S^2$, the space of unit vectors in $\mathbb{R}^3$. The constrained minimization could be written as:

$$\min\left\{ \left\| 85\mathbf{d} - \frac{\Delta t k_s n_D}{N} \sum_{i=1}^{N} \mathbf{p}_i \frac{r_{ll,i}(\mathbf{d}^*) - b_{0,i}}{m_i} \right\| : \mathbf{d}^* \in S^2 \right\} \tag{13}$$

The scalar, 85, is used to scale the target direction to the target distance of 85 mm used in the studies. Also, note that the predicted firing rates, $r_{ll,i}$ are derived from the log-linear tuning model in *Equation 12* and decoded using the tuning parameters used in the original decoder. Thus, for all target directions, $\mathbf{d}$, in the set of common targets, $\mathbb{C}$, we derived optimal re-aiming points, $\mathbf{d}^*$, which we grouped into the set, $\mathbb{D}$. The optimal re-aiming ranges were calculated as:

$$\rho_{re-aim} = \max\{\mathbf{r}_{ll}(\mathbf{d}^*) : \mathbf{d}^* \in \mathbb{D}\} - \min\{\mathbf{r}_{ll}(\mathbf{d}^*) : \mathbf{d}^* \in \mathbb{D}\} \tag{14}$$

Similarly, the tuning range predicted by the log-linear model was calculated as:

$$\rho_{ll} = \max\{\mathbf{r}_{ll}(\mathbf{d}) : \mathbf{d} \in \mathbb{C}\} - \min\{\mathbf{r}_{ll}(\mathbf{d}) : \mathbf{d} \in \mathbb{C}\} \tag{15}$$

For each unit and recording session, we calculated the sign of the difference in tuning ranges predicted by re-aiming, $\mathrm{sgn}(\rho_{re-aim,i} - \rho_{ll,i})$, and the sign of the observed difference in tuning ranges, $\mathrm{sgn}(\rho_{2D,i} - \rho_{3D,i})$, summing the results in a contingency table.

## Time course of adaptation

In order to evaluate the speed at which the neural population adapted with changes in context, we created a normalized error metric based upon tuning models from each context. In building the tuning models, we considered only the firing rates for trials where the target was in the set of common targets. For each session and context, the trials were split into two separate groups: even and odd numbered trials. We fit separate linear tuning models for each group. The error for each trial was calculated as the difference between the population response for that trial and the expected response based upon the tuning model fit from the opposite group as that trial. The errors were calculated using both the tuning models created from the 2D context and that from the 3D context. These errors were normalized by dividing by the standardized noise of the neurons from the tuning fits. The noises were estimated as the dispersion parameter from the Matlab glmfit function used to fit the tuning models. In order to standardize the normalization between the 2D and 3D contexts, we used the square root of the product of the dispersion parameters estimated from each context. The absolute value of the normalized errors were averaged over the neurons in the population. For each monkey, the trial-ordered errors were averaged across multiple recording sessions to create the final normalized error metrics displayed in *Figure 5*. For the 3D context, the errors could only be estimated for the planar targets. Subsequently, all trials for non-planar targets were removed from the analysis. The remaining planar target trials were ordered as the last planar target trial, the second to last, and so on for the purposes of averaging across recording sessions.

While plotting these normalized errors, we noticed a trend for a rapid decrease in the 2D tuning model error in the first few trials of the 2D context. In order to model that drop in the error, we fit an exponential function to the session-averaged data. The exponential function followed the equation: $y = a + b \exp(cx)$, where $y$ was the error metric and $x$ was the trial number. In our work, we use the fit parameter $c$ to describe the rapid decrease in the error metric. These fits are displayed in *Figure 5*.

## Hand-control task

In order to study whether dynamic range adaptation occurs under a hand-control paradigm, we conducted a similar experiment where the cursor movement was controlled directly via the hand of a monkey. A third monkey (Monkey N) was trained to use its hand to control the movement of a computer cursor in a center-out reaching task (*Velliste et al., 2014*). An infrared marker was placed on the back of the monkey's hand and tracked using an Optotrak 3020 optical tracking system (Northern Digital, Waterloo, Canada). The marker position was used in real time to update the position of the cursor on the stereoscopic computer monitor. The cursor and targets in the hand control paradigm were identical to those used in the brain-control study. We investigated the neural response for trials from both the 3D and 2D contexts, where the 3D context preceded the 2D context. During the trials in the 2D context, the cursor position was fixed in the *xy*-plane. In other words, the cursor would remain stationary if the monkey moved its hand closer or farther from its body while maintaining the same lateral and vertical position.

Neural firing rates were calculated in the time windows between target presentation and acquisition. These rates were averaged across trials of the same target and used to estimate the tuning ranges in each context. This analysis was limited to neurons that were directionally tuned. For each context, we fit a cosine tuning model to the firing rate responses to the set of all targets. Neurons that had a fit modulation depth less than 4 Hz in both the 2D and 3D contexts were excluded from the analysis. In *Figure 7*, we compare the difference in tuning ranges for the set of common targets between the 2D and 3D contexts in a similar manner as *Figure 2D*. This additional study and all associated procedures were performed in accordance with the guidelines of the University of Pittsburgh's Institutional Animal Care and Use Committee.

## Acknowledgements

This work was supported by NSF grant BCS1533672, the PA Department of Health Research Formula Grant SAP#4100057653 under the Commonwealth Universal Research Enhancement program, US National Institutes of Health Challenge Grant RC1 NS070311, and the Defense Advanced Research Project Agency Project N66001-12-C-4027.

## Additional information

### Funding

| Funder | Grant reference number | Author |
|---|---|---|
| National Science Foundation | BCS1533672 | Steven M Chase |
| Pennsylvania Department of Health | C.U.R.E. SAP#4100057653 | Steven M Chase |
| National Institutes of Health | Challenge Grant RC1 NS070311 | Andrew Schwartz |
| Defense Advanced Research Projects Agency | Project N66001-12-C-4027 | Andrew Schwartz |

The funders had no role in study design, data collection and interpretation, or the decision to submit the work for publication.

### Author contributions

RGR, Formal analysis, Investigation, Visualization, Methodology, Writing—original draft, Writing—review and editing; AS, Writing—original draft, Writing—review and editing; SMC, Conceptualization, Data curation, Methodology, Writing—review and editing

### Author ORCIDs

Robert G Rasmussen, http://orcid.org/0000-0002-7845-2088
Steven M Chase, http://orcid.org/0000-0003-4450-6313

### Ethics

Animal experimentation: All procedures were performed in accordance with the guidelines of the University of Pittsburgh's Institutional Animal Care and Use Committee (protocols 0508782, 14083776, and 0808279).

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
