## [Decision Letter]

Thank you for submitting your article "Dynamic range adaptation in primary motor cortical populations" for consideration by *eLife*. Your article has been reviewed by two peer reviewers, and the evaluation has been overseen by Eilon Vaadia (Reviewer #3) as the Reviewing Editor and Andrew King as the Senior Editor. The following individual involved in review of your submission has agreed to reveal their identity: Ad Aertsen (Reviewer #1).

The reviewers have discussed the reviews with one another and the Reviewing Editor has drafted this decision to help you prepare a revised submission.

Overall, we found the paper well written and addressing an interesting issue in motor neurophysiology. The experimental techniques and data analysis are sound and well described, and overall the results are valid. Additional analysis that will address the comments below may be essential.

In their manuscript, the authors set out to demonstrate that neurons in the primary motor cortex of macaques share a property that has previously been described for sensory systems, namely that, depending on context, neurons may exhibit a different firing rate range for a given action (here: a 2D vs. a 3D motor task in a virtual reality design using a BCI-decoding approach). They compare the monitored firing rates of a population of single units to the ranges predicted by three different possible scenarios: 1) a re-aiming mechanism, 2) a speed gain mechanism, 3) a dynamic range adaptation mechanism. They demonstrate that the differences between the two contexts observed are best described by a dynamic range adaptation mechanism.

The reviewers agree that they found the paper addresses an interesting issue of neuronal directional tuning in motor cortex during control of movements and its dynamic changes (under BCI, but see comments below). The authors draw an interesting conclusion, suggesting that "dynamic range adaptation" may represent a canonical feature of neural encoding of motor as well as sensory systems.

Nevertheless, the three referees have several questions, and would like to hear the authors' responses and read a revised version of the manuscript, before making a final recommendation.

Comments:

1) The question about the stability of motor encoding follows up on a somewhat obscure observation by Hepp-Reymond about the scaling of a small percentage of M1 neurons on the range of forces a monkey is asked to perform in a given context. The authors should also consider a relevant result of Humphrey DR (Science, 1970, 170(3959):758-62): When making predictions of force from multiple neurons across different loading contexts, they, "…found it necessary to scale the values of the regression coefficients in a directly proportional manner." Thus, one may wonder why the current study was performed under "BMI" conditions. We think similar observations would have greater impact if the authors addressed the normal function of the motor system. A partial answer does not come until near the end of the Discussion, when they note, in passing, "Initial, preliminary studies of a reaching version of the same task did not show the same effect as in the BCI version." The apparent difference between these two sets of results is itself important, and should affect the manner in which the BMI results are interpreted.

2) There are now a number of studies (including from this group) that address the question of the specificity with which the motor system can learn to modulate the activity of individual neurons that are used for BMI control. The typical observation is that "control" neurons tend to increase their depth of modulation and typically also change the direction of their peak tuning. It is not clear that this result is fundamentally different. These changes presumably reflect an optimal retuning of the network's activity, given the altered dynamics of the plant. The geometric arguments for a dynamic range adaptation make equally good sense in this context. The relatively weak effects for monkey C, and the fact the slope even for monkey A in Figure 4 falls well short of -1/90 (which would seem to be the prediction) further weakens the authors' argument for their specific dynamic range adaptation model. It is not clear that the depth of modulation changes represents a mechanism analogous to that of the sensory systems the authors review, or the observations of Hepp-Reymond or Humphrey. As such, the impact of the paper is somewhat reduced.

3) It is not clear that the information and manifold analyses add much. The former seems to follow pretty directly from the main result, and the latter is a questionably small effect. The authors are advised to reconsider if the information-result and Figure 6 should be introduced in the Discussion.

4) Are the authors arguing that the rather small principal angles they measure between the 2D and 3D manifolds are functionally significant, thereby requiring the monkeys to learn an "out of manifold" (see Sadtler et al.)?

5) The authors should explain better why the section on "Change in population structure" is relevant in the present context. In this context the authors are also advised to define more precisely what they mean by "the patterns of correlations that the population naturally exhibits". Are these signal correlations? Or noise correlations? Or precise spike-timing correlations? Or even something else?

6) It is stated in the Discussion that "The adaptation occurred so quickly that we could not measure its time-course. Future studies could measure the time-course of the adaptation in naive monkeys across multiple recording sessions." While appreciating this suggestion, we nevertheless feel that the authors could have found out more about the dynamics of the presumed adaptation by comparing the firing rate ranges in successive trials, after switching from one context to the next. In fact, it is hard to imagine that after the very first trial in a new context, the monkey was already able to figure out the new context. Hence, one would expect some time, that is, trials, to proceed before the monkey figures this out and, hence, is able to adjust its units' firing rate ranges. Therefore, a trial-by-trial analysis might show signs of the dynamics of the range adaptation.

7) According to Figure 1, either the 2D task (second line) or the 3D task (third line) was repeated twice, alternating with the other context. How similar were the firing rate range distributions in these repeats of the same contexts, and how dissimilar was each of these from the other context?

8) It would be interesting to hear the authors' response to the notion that the canonical mechanism is not just "dynamic range adaptation". Instead, the canonical mechanism of sensory motor computation may not be about dynamic range adaptation of a "fixed tuning", but about more general forms of additivity, based on the system state (context and more). The classical tuning of single neurons in sensory and motor areas of the brain cannot predict how a neuron is activated when the stimulus (movement) is presented (executed) in conditions that were not tested in the limited experimental condition (for example: using pure-tone or 2-D movements to test tuning of neurons etc.). It is quite possible that tuning reflects only a part of a complex network activity and may change continuously. It is possible that the dynamic range adaptation of the directional tuning is just a simple case of "tuning" changes. The authors could relate many previous examples showing that tuning in motor and sensory systems changes with learning, context, memories etc. Some non-tuned neurons can become tuned and vice versa. In sum, there may be some doubt as to whether the results and the seemingly attractive closing statement of the abstract really represent a new concept (quoting: Dynamic range adaptation "may represent a canonical feature of neural encoding of motor as well as sensory systems".)

---

## [Author Response]

[…] Nevertheless, the three referees have several questions, and would like to hear the authors' responses and read a revised version of the manuscript, before making a final recommendation.

Comments:

*1) The question about the stability of motor encoding follows up on a somewhat obscure observation by Hepp-Reymond about the scaling of a small percentage of M1 neurons on the range of forces a monkey is asked to perform in a given context. The authors should also consider a relevant result of Humphrey DR (Science, 1970, 170(3959):758-62): When making predictions of force from multiple neurons across different loading contexts, they, "…found it necessary to scale the values of the regression coefficients in a directly proportional manner." Thus, one may wonder why the current study was performed under "BMI" conditions. We think similar observations would have greater impact if the authors addressed the normal function of the motor system. A partial answer does not come until near the end of the Discussion, when they note, in passing, "Initial, preliminary studies of a reaching version of the same task did not show the same effect as in the BCI version." The apparent difference between these two sets of results is itself important, and should affect the manner in which the BMI results are interpreted.*

The Humphrey study is indeed relevant; thank you for pointing it out. We now include it as part of the motivation for this work in the Introduction. Both the Hepp-Reymond and the Humphrey studies report findings suggestive of dynamic range adaptation in motor cortex. We aimed to investigate this phenomenon further, to explore the encoding of variables, like direction, that are not monotonically represented in tuning curves. However, pursuing this work with reaching movements is complicated by additional external and environmental factors such as gravity, inertia, etc. (It is because of these additional factors that it was inaccurate for us to describe the hand control version of this as the “same task”, and we have reworded that sentence to describe it as a “similar task.”) The BCI paradigm allowed us to circumvent these complicating factors, which is the main reason we employed it here. However, we agree that the difference between the BCI and hand-control results are interesting and important. We now include data from a hand-control version of this task in a new figure (Figure 7) at the end of the Results section. We do not see evidence of dynamic range adaptation in the hand-control data. We speculate on reasons for this in the Discussion section. The text has been changed as follows to reflect these additions:

From Results:

**“**No Adaptation with Hand-Control Task

BCI provides a powerful framework for addressing basic scientific questions about motor control, including the dynamic range adaptation effects observed here, because the effect each neuron has on cursor movement is fully specified by the experimenter, and because we can obviate complicating factors such as gravity and inertia. […] Unlike with the BCI task, we see no evidence of dynamic range adaptation in the hand-control version of this task. We speculate on the reasons for this in the Discussion section.”

From Discussion:

“While we observed consistent dynamic range adaptation between the 2D and 3D contexts in the BCI task, we did not observe it in the hand-control version of this task. […] If this is true, it suggests that isometric force production is a closer analog to BCI than the overt reaching movements to which it is typically compared.”

2) There are now a number of studies (including from this group) that address the question of the specificity with which the motor system can learn to modulate the activity of individual neurons that are used for BMI control. The typical observation is that "control" neurons tend to increase their depth of modulation and typically also change the direction of their peak tuning. It is not clear that this result is fundamentally different. These changes presumably reflect an optimal retuning of the network's activity, given the altered dynamics of the plant. The geometric arguments for a dynamic range adaptation make equally good sense in this context. The relatively weak effects for monkey C, and the fact the slope even for monkey A in Figure 4 falls well short of -1/90 (which would seem to be the prediction) further weakens the authors' argument for their specific dynamic range adaptation model. It is not clear that the depth of modulation changes represents a mechanism analogous to that of the sensory systems the authors review, or the observations of Hepp-Reymond or Humphrey. As such, the impact of the paper is somewhat reduced.

We appreciate the opportunity to clarify our contribution. A major difference between our study and previous BMI studies that have demonstrated changes in the tuning of individual neurons (e.g., Jarosiewicz et al., 2008, Ganguly and Carmena, 2009, Ganguly and Carmena, 2011, or Chase et al., 2012) is that in our study, there was no overt learning pressure. In each of those previous studies, the subject started with a “perturbed” BMI mapping that impaired control. In contrast, in this study there was no sudden impairment when moving from the 3D to the 2D context, and had subjects continued doing what they had been doing in the 3D context, there would have been no overall change in performance. Despite the lack of learning pressure, we see changes in the dynamic range of tuning when moving between 3D and 2D. Another major difference is the time course of neural tuning changes. The time course of neural tuning changes in response to altered plant dynamics is on the order of tens of minutes (Jarosiewicz et al.) to days (Ganguly and Carmena). In this study, we observed changes in neural tuning that occurred within seconds. We have updated the Discussion to clarify these points:

“Other studies have noted tuning changes of individual neurons during BCI learning tasks (Jarosiewicz et al., 2008; Ganguly and Carmena, 2009; Ganguly and Carmena, 2011; Chase et al., 2012). However, the dynamic range adaptation we see appears to be fundamentally different from the effects shown in those studies. First, the time scale of dynamic range adaptation is on the order of seconds (Figure 5). The time scale of BCI-learning-related changes in neural tuning is on the order of tens of minutes (Jarosiewicz et al., 2008; Chase et al., 2012) to days (Ganguly and Carmena, 2009; Ganguly and Carmena, 2011). Second, the dynamic range adaptation is restricted to changes in tuning depth; changes in preferred direction are minimal. In BCI learning studies, preferred directions typically change as well. Third, dynamic range adaptation occurs without any overt learning pressure. In studies of BCI learning, the mapping between neural activity and cursor movements is changed in such a way to impair task performance. That was not the case here: if the subject continued using the same firing rates in the 3D and 2D contexts, performance would have been unaffected. For these reasons, we speculate that dynamic range adaptation represents an additional degree of flexibility in neural tuning in the motor system.”

As to the comment about the completeness of the effect, Figure 4 is not ideally suited to address this point. This is principally because the ratio on the ordinate axis will never be negative. Thus, noise in the measurement of the dynamic range will bias the curve such that it appears the effect is incomplete. Instead, to assess completeness we developed a new metric, whose results we display in a new Figure 6 in the Results section:

“We also assessed the completeness of the dynamic range adaptation. To be complete, the dynamic range in the 2D context should equal the full dynamic range in the 3D context, when all targets are taken into account. […] Ohzawa developed an adaptability index to characterize the contrast gain adaptation of cells in visual cortex, and reported values in line with our results (cf. Figure 5 of Ohzawa et al., 1985).”

3) It is not clear that the information and manifold analyses add much. The former seems to follow pretty directly from the main result, and the latter is a questionably small effect. The authors are advised to reconsider if the information-result and Figure 6 should be introduced in the Discussion.

On reflection, we agree. We have removed these analyses from the paper.

4) Are the authors arguing that the rather small principal angles they measure between the 2D and 3D manifolds are functionally significant, thereby requiring the monkeys to learn an "out of manifold" (see Sadtler et al.)?

Our attempt with the principal angle analysis was to determine if the dynamic range adaptation effects that we see “warp” the manifold in a way that would make it look as if the 2D control was out of manifold. Our interpretation would not be that such control was hard to learn because it’s out of manifold, as found in Sadtler et al. In fact, we find the opposite: subjects have no trouble switching between 3D and 2D contexts. Rather, our interpretation would be that using a 3D-only task, we get an incomplete characterization of the manifold because of the dynamic range adaptation that neurons can undergo. However, we agree that the results were relatively small and the point is a bit obscure. We have removed this analysis and discussion from the paper.

5) The authors should explain better why the section on "Change in population structure" is relevant in the present context. In this context the authors are also advised to define more precisely what they mean by "the patterns of correlations that the population naturally exhibits". Are these signal correlations? Or noise correlations? Or precise spike-timing correlations? Or even something else?

The text referred to here was also meant to make the point about the consequences of manifold misestimation: using only a 3D task, the estimated manifold (that is predictive of BCI learnability, Sadtler et al.) may be incomplete. Thus, one might mistakenly think that 2D control would be hard to learn. As the reviewer points out, we were not clear in our choice of language. We have removed this analysis and discussion from the paper.

6) It is stated in the Discussion that "The adaptation occurred so quickly that we could not measure its time-course. Future studies could measure the time-course of the adaptation in naive monkeys across multiple recording sessions." While appreciating this suggestion, we nevertheless feel that the authors could have found out more about the dynamics of the presumed adaptation by comparing the firing rate ranges in successive trials, after switching from one context to the next. In fact, it is hard to imagine that after the very first trial in a new context, the monkey was already able to figure out the new context. Hence, one would expect some time, that is, trials, to proceed before the monkey figures this out and, hence, is able to adjust its units' firing rate ranges. Therefore, a trial-by-trial analysis might show signs of the dynamics of the range adaptation.

We have included a new trial-by-trial analysis of the dynamics of the range adaptation. This new analysis is explained more thoroughly in the revised manuscript. Briefly, we assessed how quickly neurons switched their tuning from their 3D tuning curves to their 2D tuning curves. By analyzing the model fit error over the transition from 3D to 2D, we were able to assess the time course of dynamic range adaptation. This analysis has been added to the Results section and included as a new Figure 5.

“The adaptation process appeared to occur rapidly with the switch from the 3D to 2D context. In order to investigate the timecourse of this process, we devised a method to estimate changes in dynamic range on a trial-by-trial basis (see Methods). […] The 2D model error is immediately lower than the 3D model error on the first 2D trial and rapidly reduces to a baseline state within the first few trials.”

7) According to Figure 1, either the 2D task (second line) or the 3D task (third line) was repeated twice, alternating with the other context. How similar were the firing rate range distributions in these repeats of the same contexts, and how dissimilar was each of these from the other context?

This is an important control. We have compared the tuning changes between repeats of the same task to the tuning changes we see from 3D to 2D. We have included this analysis in Figure 2—figure supplement 5. We have added the following details of this analysis to the Results section:

“As shown in Figure 1, there was a subset of recording sessions where a task context was repeated after the alternative context. […] This suggests that our results from Figure 2 may be a slight underestimate of the actual tuning increase.”

8) It would be interesting to hear the authors' response to the notion that the canonical mechanism is not just "dynamic range adaptation". Instead, the canonical mechanism of sensory motor computation may not be about dynamic range adaptation of a "fixed tuning", but about more general forms of additivity, based on the system state (context and more). The classical tuning of single neurons in sensory and motor areas of the brain cannot predict how a neuron is activated when the stimulus (movement) is presented (executed) in conditions that were not tested in the limited experimental condition (for example: using pure-tone or 2-D movements to test tuning of neurons etc.). It is quite possible that tuning reflects only a part of a complex network activity and may change continuously. It is possible that the dynamic range adaptation of the directional tuning is just a simple case of "tuning" changes. The authors could relate many previous examples showing that tuning in motor and sensory systems changes with learning, context, memories etc. Some non-tuned neurons can become tuned and vice versa. In sum, there may be some doubt as to whether the results and the seemingly attractive closing statement of the abstract really represent a new concept (quoting: Dynamic range adaptation "may represent a canonical feature of neural encoding of motor as well as sensory systems".)

In general, we agree with the idea that directional tuning is only one aspect of the control process taking place during volitional movement. We adopt the view that by recording neural activity, we are monitoring information as it flows through the node (neuron) that we happen to be monitoring. This is quite distinct from the quest to determine the causal contribution of a single neuron to movement generation. We agree that neural tuning is flexible, and can change with learning, experience, or a number of other factors. However, dynamic range adaptation helps bring some order out of the chaos of tuning flexibility. In this set of experiments, it is not the case that we have two sets of neural tunings that are completely unrelated to one another. Rather, they are related by an information processing principle that enables the tuning in the 2D context to be predicted from the tuning in the 3D context, and vice versa, at least up to the preferred tuning in depth. This is similar to what has been consistently observed in sensory systems, which is why we term it a “canonical feature of neural encoding.” We certainly do not mean to imply that it is the only mechanism of tuning curve change. However, it is one of the few instances in the motor system in which the changes in tuning of individual neurons can actually be predicted a priori.

We have added the following to the Discussion to address this question:

“Direction as an important component of the control signal is undeniable. Although directional tuning has been shown to change in a context-dependent manner, it should be emphasized that directional tuning is stable across repetitions within the same condition. […] By better defining which aspects of behavior lead to tuning changes and from more complete descriptions of the factors driving these changes, we will gain a deeper understanding of the neural operations taking place during the control of movement.”